# Metabolic Profile of Patients with Smith-Magenis Syndrome: An Observational Study with Literature Review

**DOI:** 10.3390/children10091451

**Published:** 2023-08-25

**Authors:** Clelia Cipolla, Linda Sessa, Giulia Rotunno, Giorgio Sodero, Francesco Proli, Chiara Veredice, Valentina Giorgio, Chiara Leoni, Jessica Rosati, Domenico Limongelli, Eliza Kuczynska, Elisabetta Sforza, Valentina Trevisan, Donato Rigante, Giuseppe Zampino, Roberta Onesimo

**Affiliations:** 1Pediatric Unit, Department of Life Sciences and Public Health, Fondazione Policlinico Universitario A. Gemelli IRCCS, 00168 Rome, RM, Italy; clelia.cipolla@policlinicogemelli.it (C.C.); giuseppe.zampino@policlinicogemelli.it (G.Z.); roberta.onesimo@policlinicogemelli.it (R.O.); 2Università Cattolica Sacro Cuore, 00168 Rome, RM, Italy; 3Pediatric Neurology Unit, Fondazione Policlinico Universitario A. Gemelli, IRCCS, 00168 Rome, RM, Italy; 4Center for Rare Diseases and Birth Defect, Fondazione Policlinico Universitario A. Gemelli IRCCS, 00168 Rome, RM, Italy; 5Unità di Riprogrammazione Cellulare, Fondazione IRCCS Casa Sollievo della Sofferenza, 71013 San Giovanni Rotondo, FG, Italy

**Keywords:** Smith-Magenis syndrome, obesity, hypercholesterolemia, lipids, insulin, diabetes, metabolic profile, nutrition, personalized medicine

## Abstract

*Background*: Smith-Magenis syndrome (SMS) is caused by either interstitial deletions in the 17p11.2 region or pathogenic variants in the *RAI1* gene and is marked by a distinct set of physical, developmental, neurological, and behavioral features. Hypercholesterolemia has been described in SMS, and obesity is also commonly found. *Aim*: To describe and characterize the metabolic phenotype of a cohort of SMS patients with an age range of 2.9–32.4 years and to evaluate any correlations between their body mass index and serum lipids, glycated hemoglobin (HbA1c), and basal insulin levels. *Results*: Seven/thirty-five patients had high values of both total cholesterol and low-density lipoprotein cholesterol; 3/35 had high values of triglycerides; none of the patients with *RAI1* variants presented dyslipidemia. No patients had abnormal fasting glucose levels. Three/thirty-five patients had HbA1c in the prediabetes range. Ten/twenty-two patients with 17p11.2 deletion and 2/3 with *RAI1* variants had increased insulin basal levels. Three/twenty-three patients with the 17p11.2 deletion had prediabetes. *Conclusion*: Our investigation suggests that SMS ‘deleted’ patients may show a dyslipidemic pattern, while SMS ‘mutated’ patients are more likely to develop early-onset obesity along with hyperinsulinism.

## 1. Introduction

Smith-Magenis syndrome (SMS) is a genetic disorder caused by both heterozygous deletions at the chromosome band 17p11.2 region or heterozygous pathogenic variants in the Retinoic Acid-Induced 1 (*RAI1*) gene, which is characterized by multiple congenital anomalies, sleep disturbances, behavioral impairment, and a variable level of intellectual disability [1]. The majority of 17p11.2 deletions are de novo (90%), while deleterious point mutations of *RAI1* (10%) can be inherited [2]. Even though the SMS phenotype is usually not evident before 18 months of age, either failure to thrive or remarkable hypotonia might be present since birth [3]. Distinctive clinical features include brachycephaly with a broad face, wide nasal bridge, prominent forehead, prognathism with a protruding premaxilla, brachydactyly, a “tented” vermilion of the upper lip, and major renal and brain abnormalities such as ventriculomegaly [4]. Cardiac involvement (atrial and interventricular defects, valvular stenosis, and major cardiac anomalies) is commonly seen in patients with 17p11.2 deletions, while it is seldom encountered in cases caused by *RAI1* variants [1,5]. Neurocognitive impairment usually includes mild-to-moderate intellectual disability, stereotypies like self-hugging, self-destructive behaviors like self-biting and hitting, onychotillomania, and polyembolokoilamania [6]. Hormonal secretion of melatonin is often compromised in SMS, leading to increased production during the day and a relative deficiency at night. This alteration causes a disruption of the circadian rhythm and severely impaired quality of life [7].

Obesity is also common in SMS patients, depending on a multifactorial etiology that might involve hyperphagic behaviors and sedentary lifestyles [8,9]. Blood lipid abnormalities such as hypercholesterolemia are also frequently observed [10] and could contribute to an increased cardiovascular risk in SMS patients. The general metabolic implications of SMS are poorly known, and only partial data from individual patients or small cohorts of patients have been reported in the medical literature [11].

The main objective of our study was to analyze the metabolic characteristics of a cohort of Italian patients with SMS and evaluate a possible correlation between different blood tests and body mass index (BMI). Specifically, total lipids, glucose, insulin, homeostatic model assessment for insulin resistance (HOMA-IR), glycated hemoglobin (HbA1c), thyroid stimulating hormone [TSH], fT3, and fT4 were assessed. In addition, a review of the medical literature related to SMS and metabolic issues has been performed without restriction on the publication dates.

## 2. Patients and Methods

We conducted a single-center observational study investigating alterations in the metabolic profile of Italian patients with a confirmed genetic diagnosis of SMS. The Local Ethical Committee in our University approved the study as part of a larger protocol based on the evaluation of disability and nutritional aspects in patients with rare diseases (approval code from our Ethical Committee: 2105; approval date: 5 February 2019). 

In particular, we retrospectively analyzed the clinical records of patients with molecularly confirmed SMS who were on regular follow-up for at least one year at the Rare Disease Unit of the Fondazione Policlinico Universitario A. Gemelli IRCCS during the period September 2019-December 2022. No age limits were set. Our initial screening process found a total of 39 potentially recruitable patients. Of them, 4 subjects were excluded because of lack of laboratory data (n = 1), short follow-up period (n = 2), or transfer to another center (n = 1). The remaining 35 unrelated patients with SMS were 20 females (57%) and 15 males (43%); they had a mean age of 13.9 ± 8 years (age range: 2.9-32.4 years); none of them were using any medications that might affect lipid levels or glucose metabolism.

### 2.1. Data Extraction

As a first step, accurate data on genetic findings were recorded in order to further explore a genotype-phenotype correlation. Data on patients’ auxological parameters (age, weight, height, BMI) and metabolic parameters (laboratory tests including total cholesterol, low-density lipoprotein [LDL] cholesterol, high-density lipoprotein [HDL] cholesterol, triglycerides, fasting glucose, insulin, Hb1Ac, TSH, fT3, and fT4) were extracted. When possible, we also calculated for each patient the HOMA-IR index [12], an indirect marker of insulin resistance derived from basal blood tests (HOMA-IR = fasting glucose levels [mmol/L] × fasting insulin levels [μU/mL]/22.5 or fasting glucose levels [mg/dL] × fasting insulin levels [μU/mL]/405).

### 2.2. Data Interpretation

Auxological parameters were studied after calculating the reference percentiles and standard deviations (SD) for age and biological sex of each patient to obtain homogeneous and comparable data. BMI percentiles were divided into four categories according to international growth charts [13]: *underweight* (less than the 5th percentile), *healthy weight* (from the 5th to the 85th percentile), *overweight* (between the 85th and 95th percentile), and *obesity* (equal to or over the 95th percentile). According to the Expert Panel on Integrated Guidelines for Cardiovascular Health and Risk Reduction in Children and Adolescents [14], we divided plasma lipid concentrations (mg/dL) into the following categories:−Total cholesterol: *acceptable* < 170, *borderline* 170–199, *high* ≥ 200;−LDL-cholesterol: *acceptable* < 110, *borderline* 110–129, *high* ≥ 130.−HDL-cholesterol: *acceptable* > 45, *borderline* 40–45, *low* < 40.

In agreement with the Italian Society of Diabetology [15], we considered fasting glucose values ≤ 100 mg/dL as normal. HbA1c was considered normal if <39 mmol/mol, diagnostic of prediabetes if 39–46 mmol/mol, and suggesting diabetes mellitus if ≥48 mmol/mol. Insulin was interpreted as normal or increased (>10 microUI/L), as usually considered in our center. HOMA-IR was calculated using the previously shown formula: the HOMA-IR cut-off point to differentiate patients with low or high HOMA-IR was 2.5, as for adults [16].

### 2.3. Statistical Analysis

Categorical variables were reported as numbers; continuous variables were reported as means and SD. For the continuous variables, the Shapiro test was used to assess whether the distribution was normal or not. Considering that not all continuous variables had a normal distribution, the Spearman correlation test was performed to highlight whether there were any parameters related to BMI (evaluated as BMI in absolute values, BMI percentiles, or BMI percentiles divided into classes: *underweight*: less than the 5th percentile; *healthy weight*: 5th to less than the 85th percentile; *overweight*: 85th to less than the 95th percentile; *obesity*: equal to or over the 95th percentile). The non-parametric Mann-Whitney U test was performed to assess if there were significant differences in terms of metabolic variables between patients with low and high BMI percentiles (cut-off: 85th percentile). In addition, a Mann-Whitney U test was performed to determine whether metabolic variables were statistically different between males and females and between younger and older patients (cut-off: 10 years); the age of 10 was chosen because this is the most frequent period at which most SMS patients tend to become overweight or frankly obese [3].

Statistical analysis was performed using the IBM SPSS Statistics 25.0 software (IBM Corporation, Armonk, NY, USA). In all cases, statistical significance was set when alpha was <0.05.

## 3. Results

### 3.1. Characteristics of Our Cohort of Patients with SMS

Data on height, weight, BMI, lipid profile, and fasting glucose were available for all patients. Twenty-nine patients (82.8%) were also evaluated for thyroid function, 26 (74.2%) for HbA1c, and 25 (71.4%) for basal insulin level. The overall characteristics of our cohort of SMS children are listed in Table 1. More specifically, 86% of SMS patients (30/35) had heterozygous deletions at the 17p11.2 region (including *RAI1*), while 5 SMS patients (14%) had *RAI1* pathogenic variants (as detailed in Table 2).

In our cohort, 43% of patients (15/35) were classified as *obese*, 11% (4/35) as *overweight*, and 46% (16/35) had a *healthy weight*. Twenty percent of subjects (7/35) had high values of both total cholesterol and LDL cholesterol, while 63% (22/35) had low HDL cholesterol levels, and 8.6% (3/35) had high values of triglycerides. None of the patients with RAI1 variants had dyslipidemia. None of our patients had abnormal fasting glucose levels (>100 mg/dL). For nine patients (25.7%), HbA1c data were missing; 23/26 patients (88%) had normal HbA1c, while 3/26 (11.5%) had values in the prediabetes range. Basal insulin data were not available for 10 subjects; among the other 25 patients, 12/25 (48%) had insulin levels > 10 microUI/L, while 10/25 (40%) had HOMA-IR values > 2.5 (see Table 3). Thyroid function was normal in all patients studied (29/35).

### 3.2. Correlation Analysis

There was no statistically significant correlation between BMI percentiles and total cholesterol, LDL cholesterol, triglycerides, glucose, insulin, HbA1c, or HOMA-IR. A strong negative correlation was found between BMI percentiles and HDL (rho = −0.749, *p* < 0.001). A mild positive correlation was found between BMI (absolute values) and triglycerides (rho = 0.392, *p* = 0.020) and between BMI percentiles and HbA1c (rho = 0.524, *p* = 0.006). A mild positive correlation was also found between triglycerides and age (rho = 0.489, *p* = 0.003), cholesterol (rho = 0.432, *p* = 0.003), insulin (absolute values) (rho = 0.466, *p* = 0.019), HOMA-IR (rho = 0.497, *p* = 0.012), and HbA1c (rho = 0.413, *p* = 0.036). A mild positive correlation was found between insulin and age (rho = 0.460, *p* = 0.021) and insulin and triglycerides (rho = 0.479, *p* = 0.015). A positive mild correlation was also found between HbA1c and BMI percentiles (rho = 0.524, *p* = 0.006), while a negative mild correlation was found between HbA1c and HDL (rho = −0.394, *p* = 0.046).

### 3.3. Application of the Mann Whitney U-Test

No statistically significant differences were found for all the variables studied, except for HDL cholesterol (U = 30, *p* < 0.001), showing that lower levels of HDL were associated with higher BMI percentiles (Figure 1). No statistically significant differences were found for all the variables or patients’ sex. Moreover, statistically significant differences were found for triglycerides (U = 221, *p* = 0.017), insulin (U = 128, *p* = 0.004), and HOMA-IR (U = 129, *p* = 0.003), which were higher in older patients (Figure 2).

### 3.4. Summary of the Results

The statistical analysis highlighted no significant correlation between BMI percentiles and total cholesterol, LDL cholesterol, or triglycerides. A strong negative correlation was found between BMI percentiles and HDL. A positive mild correlation was found between BMI and triglycerides; a positive mild correlation was found between HbA1c and BMI percentiles; no statistical differences were found for all the variables studied in patients with lower and higher BMI percentiles, except for HDL, showing that lower HDL can be found in patients with higher BMI percentiles.

### 3.5. Review of Medical Literature

A review of the medical literature related to SMS and metabolic issues was performed following the Narrative Review Reporting Checklist (see Appendix A). Three databases were analyzed without restriction about the publication dates. English full-text records were included after matching the terms “Smith-Magenis syndrome” and “metabolic” or “lipid metabolism” (Appendix A). Two independent physicians performed the search. Selected documents were cross-checked, and duplicates were removed. The literature search yielded all potentially useful articles: 72 ‘full-text’ manuscripts were retrieved; of them, 16 articles fully met the inclusion criteria. The articles recruited ranged from 1999 to 2023, spanning a 25-year period.

## 4. Discussion

Since the first report of *RAI1* haploinsufficiency syndrome in 1986, little is known about the metabolic implications of SMS, which is marked by the occurrence of early-onset obesity. Metabolic data related to these patients are scanty, and—most of all—it is unclear whether these alterations are due to the underlying genetic abnormalities or to erratic eating behaviors [10]. The relationship between different genetic disorders and the processing and distribution of macronutrients such as lipids and carbohydrates is many-sided and follows different mechanisms. For instance, mutations in several genes causing developmental disorders like Rett syndrome may directly result in perturbed lipid metabolism [17]. The pathogenesis of specific rare diseases such as Smith-Lemli-Opitz syndrome and mevalonate kinase deficiency may arise from enzymatic defects in components of the sterol biosynthetic pathway, resulting in a shortage of downstream products [18,19]. Additionally, a significant increase in plasma levels of triglycerides may reflect the risk of macrophage activation syndrome occurring in protean clinical settings [20].

Particularly for SMS, a few medical reports have indicated that alterations of both basal glucose and insulin can be frequently found with a potentially increased risk of developing diabetes mellitus type 2, insulin resistance, and metabolic syndrome during adulthood [3]. Higher levels of total cholesterol were reported for SMS patients by Valerio et al. [15], but to the best of our knowledge, the present study is one of the first to analyze a wider metabolic profile in a cohort of patients with SMS.

In our center, all patients with obesity-risk syndromes undergo routine blood tests to assess the presence of any potential metabolic impairment. We conducted a monocentric retrospective study evaluating the laboratory characteristics of 35 subjects with SMS, including total cholesterol, LDL cholesterol, HDL cholesterol, triglycerides, fasting glucose, Hb1Ac, and insulin. Patients carrying the deletion showed a distinctive hypercholesterolemic footprint: 20% of them had elevated total and LDL cholesterol values, 63% had low HDL cholesterol, and 8% had elevated triglycerides. Conversely, none of the patients with *RAI1* variants had elevated cholesterol or triglyceride levels. This is the first time that such a point has arisen in the literature, as the main study dealing with metabolic issues in SMS was published before the causative *RAI1* variants were identified [10,21]. Indeed, Smith et al. found hypercholesterolemia also in children with low or normal BMI, concluding that BMI by itself cannot explain the rising trend of total cholesterol, LDL cholesterol, and/or triglycerides in SMS [10].

The relationship between SMS genetic alterations and metabolic dysregulation has not been clarified yet. Lacaria et al. found that 17p11.2-deleted mice had increased weight, higher body fat mass, decreased HDL levels, and even reduced insulin sensitivity compared with mice with duplication of the same region [22].

A difference between mutated and deleted patients also emerged in the lipid profile. Turco et al. generated and characterized primary cells derived from four SMS patients (two with the deletion and two carrying *RAI1* variants) and four control subjects to investigate the molecular processes underlying SMS. By combining transcriptomic and lipidomic analyses, they found that SMS patients had altered expression of regulatory genes involved in lipid metabolism [23]. There are different hypotheses regarding hypercholesterolemia and a possible dysregulation in the molecular pathway of SMS [24]. A role for genes localized within the SMS region, such as the *SREBP1* (sterol regulatory element binding protein-1) gene, has been hypothesized [25]: it encodes a transmembrane transcription factor that activates cholesterol transport, endogenous cholesterol synthesis, and LDL receptor expression [26]. Shimano et al. also studied this possible association in an animal model, finding that heterozygous and homozygous mice with targeted disruptions of *SREBP1* had higher levels of mRNA involved in cholesterol synthesis [27]. Despite these findings, which could justify the relationship between *SREBP1* haploinsufficiency and hypercholesterolemia in SMS patients, there is not sufficient evidence to further support this hypothesis.

From our analysis, no patients with *RAI1* variants had dyslipidemia, and a strong negative correlation was found between BMI percentiles and HDL. Moreover, we found no statistical differences in glucose levels between patients with lower or higher BMI percentiles, though we confirmed a mild positive correlation between HbA1c and BMI. More than half of our population had glucose or insulin alterations: 48% of patients had insulin levels > 10 microUI/L, while 11% had HbA1c in the prediabetes range.

Many studies have clearly shown that among children and adolescents with overweight and obesity, particularly those with central fat distribution [28], levels of insulin may be increased and dyslipidemia may occur [29]. Insulin resistance is also common and a predictor of developing diabetes mellitus type 2 during adulthood (7–25% of obese adolescents have an increased risk of developing diabetes) [30]. In our cohort, half of SMS patients were obese, and it is well-established that hyperphagia and a general sedentary lifestyle might predispose to obesity in SMS [6]. Results of previous studies suggested that boys with SMS may be more overweight and exhibit more severe eating behaviors than girls with SMS. From our analysis, no gender difference emerged. Finally, we also know that SMS patients with *RAI1* variants may have a higher propensity to obesity [31,32], but our results indicate a risk of developing higher insulin concentrations in mutated SMS patients than in those with a deletion of the 17p11.2 region. Furthermore, 13% of deleted patients have shown abnormal results for Hb1Ac. Further studies investigating whether other genes might affect glucose metabolism in SMS are needed.

Despite the general results of our study, some limitations need to be declared as well: first of all, the retrospective nature of this analysis, but also the lack of a control group population and a relatively limited number of SMS patients (due to the monocentric structure of our protocol).

## 5. Conclusions

Even if the lack of a control group population does not allow for general assumptions, for the first time our data reveal relevant insights into the metabolic profiling of SMS patients, including patients with *RAI1* variants. In particular, our analysis highlights that SMS ‘deleted’ patients may show a dyslipidemic pattern, while ‘mutated’ patients carrying *RAI1* variants are more likely to develop early-onset obesity along with hyperinsulinemic dysregulation.

## Figures and Tables

**Figure 1 children-10-01451-f001:**
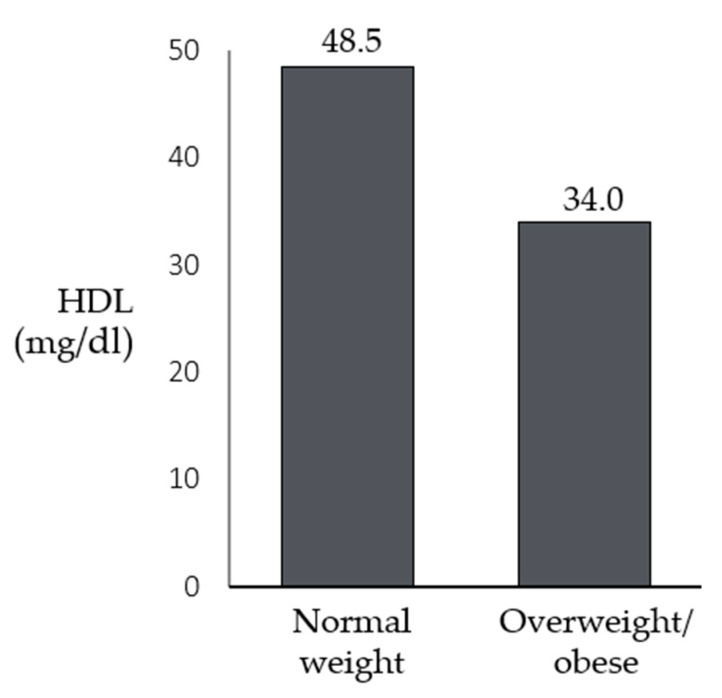
Differences in the medians of HDL between patients with Smith-Magenis syndrome who have a normal weight and those who are overweight/obese.

**Figure 2 children-10-01451-f002:**
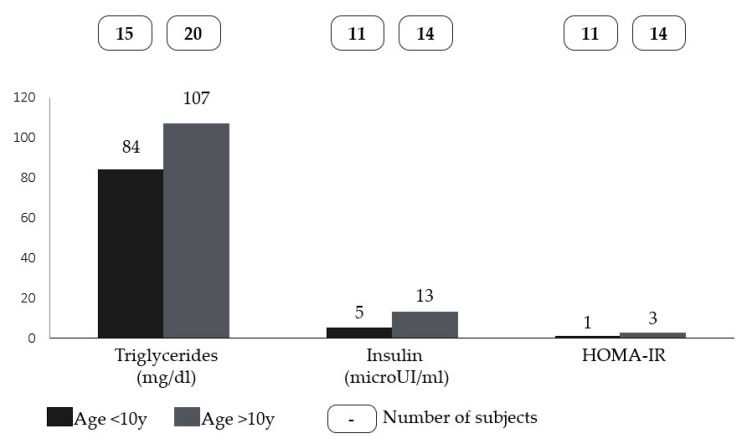
Differences in the medians of triglycerides, insulin, and HOMA-IR between younger and older patients with Smith-Magenis syndrome (cut-off set at 10 years).

**Table 1 children-10-01451-t001:** Clinical and metabolic characteristics of our population of patients with Smith-Magenis syndrome enrolled in the study.

	Mean ± SD (Max-Min)	N. of Subjects (Total: 35)
Age (years)	13.9 ± 8 (32.4–2.9)	35
Body mass index (kg/m^2^)	25.1 ± 9.2 (58.2–14.2)	35
Body mass index (percentiles)	79.9 ± 24.6 (100–10)	35
Cholesterol (mg/dL)	167.5 ± 32.9 (244–116)	35
LDL (mg/dL)	105.2 ± 25.5 (165–60)	34
HDL (mg/dL)	40.4 ± 10.2 (64–23)	35
Triglycerides (mg/dL)	108.11 ± 66.9 (461–46)	35
Glucose (mg/dL)	80.83 ± 8.5 (99–62)	35
Insulin (microUI/mL)	10.5 ± 7 (28.5–1.4)	25
HOMA-IR	2.1 ± 1.4 (5.4–0.32)	25
HbA1c (mmol/L)	34 ± 4.4 (44–22)	26
TSH (microUI/mL)	1.9 ± 0.9 (3.9–0.69)	29
fT3 (pg/mL)	4 ± 0.6 (5.3–3.1)	29
fT4 (ng/mL)	10.8 ± 1.3 (13.6–8.9)	29

**Table 2 children-10-01451-t002:** Metabolic alterations of patients with Smith-Magenis syndrome enrolled in the study in association with the genetic alteration (deletion of 17p11.2—detected by array CGH or FISH—and *RAI1* variants).

Patients	Genetic Test	Result	Total Cholesterol >200 mg/dL	Triglycerides >150 mg/dL	Insulin >10 microUI/L	Hb1Ac ≥39 mmol/mol	CDC Class
1	CGH array	del 17p11.2 (3.9 Mb)	No	No	No	No	normal weight
2	CGH array	del 17p11.2 (3.3 Mb)	Yes	No	-	-	normal weight
3	CGH array	del 17p11.2 (3.7 Mb)	No	No	No	No	obese
4	CGH array	del 17p11.2 (3.6 Mb)	No	No	Yes	No	normal weight
5	CGH array	del 17p11.2 (1.2 Mb)	No	No	No	Yes	obese
6	CGH array	del 17p11.2 (1.5 Mb)	No	No	No	No	normal weight
7	CGH array	del 17p11.2 (4.6 Mb)	Yes	Yes	Yes	No	normal weight
8	CGH array	del 17p11.2 (1 Mb)	No	No	No	No	overweight
9	CGH array	del 17p11.2 (4 Mb)	No	No	Yes	No	obese
10	CGH array	del 17p11.2 (4.3 Mb)	No	No	No	No	normal weight
11	CGH array	del 17p11.2 (3.6 Mb)	No	No	Yes	No	obese
12	CGH array	del 17p11.2 (3.3 Mb)	No	No	Yes	No	normal weight
13	CGH array	del 17p11.2 (3.8 Mb)	Yes	No	Yes	No	normal weight
14	CGH array	del 17p11.2 (3.6 Mb)	No	No	No	No	normal weight
15	CGH array	del 17p11.2 (4.7 Mb)	Yes	No	-	-	normal weight
16	CGH array	del 17p11.2 (3 Mb)	No	No	-	No	normal weight
17	CGH array	del 17p11.2 (9 Mb)	No	No	Yes	Yes	obese
18	CGH array	del 17p11.2 (3.3 Mb)	Yes	No	-	-	normal weight
19	CGH array	del 17p11.2 (3.7 Mb)	No	No	No	No	overweight
20	CGH array	del 17p11.2 (3.7 Mb)	No	Yes	Yes	-	normal weight
21	CGH array	del 17p11.2 (3.3 Mb)	No	No	-	-	normal weight
22	CGH array	del 17p11.2 (3.7 Mb)	No	Yes	Yes	Yes	obese
23	CGH array	del 17p11.2 (2.1 Mb)	No	No	-	-	obese
24	CGH array	del 17p11.2 (3.4 Mb)	No	No	-	-	obese
25	FISH	del 17p11.2	Yes	No	-	No	obese
26	FISH	del 17p11.2	No	No	No	No	obese
27	FISH	del 17p11.2	No	No	No	No	obese
28	FISH	del 17p11.2	Yes	No	-	-	
29	FISH	del 17p11.2	No	No	No	No	obese
30	FISH	del 17p11.2	No	No	Yes	No	obese
31	Sequencing	*RAI1* variant c.3828delC p.Lys1277Argfs*38	No	No	Yes	No	obese
32	Sequencing	*RAI1* variant c.373C>T p.Gln125*	No	No	Yes	No	obese
33	Sequencing	*RAI1* variant c.2969_2970delAG p. Glu990Alafs*38	No	No	-	No	normal weight
34	Sequencing	*RAI1* variant c.3828delC p.Lys1277Argfs*38	No	No	Yes	No	obese
35	Sequencing	*RAI1* variant c.373C>T p.Gln125*	No	No	Yes	No	obese

**Table 3 children-10-01451-t003:** Patients with Smith-Magenis syndrome enrolled in our study divided according to the genetic abnormality in relationship with CDC class of overweight/obesity and results of the main metabolic parameters.

	17p11.2 Deletion	*RAI1* Mutation	Total
Overweight/obesity	16/30 *(53%)*	3/5 *(60%)*	19/35 *(54%)*
Total cholesterol > 200 mg/dL	7/30 *(23%)*	0/5	7/35 *(20%)*
Triglycerides > 150 mg/dL	3/30 *(10%)*	0/5	3/35 *(8.6%)*
Insulin > 10 microUI/L	10/22 *(45%)*	2/3 *(66%)*	12/25 *(48%)*
Hb1Ac ≥ 39 mmol/mol	3/23 *(13%)*	0/3	3/26 *(11.5%)*

## Data Availability

No datasets were generated and analyzed during the study.

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
