# Peer review of "Metabolic Profile of Patients with Smith-Magenis Syndrome: An Observational Study with Literature Review"

_children, 2023, doi:10.3390/children10091451_

Round 1

Reviewer 1 Report

The authors present some laboratory parameters of a relatively small Smith-Magenis syndrome (SMS) cohort (35 patients). SMS is caused by 17p11.2 deletions, encompassing multiple genes including retinoic acid-induced 1 gene (RAI1), or by mutations in the RAI1 gene itself. The author’s key conclusion is that patients with 17p11.2 deletions (n=30) are prone to develop dyslipidemia while patients with RAI1 mutations (n=5) tend to develop signs of hyperinsulinemia. The following points (in order of appearence) need attention:

Results: lines 148-166 use of integers vs. decimal numbers?

ALT activity mentioned in line 149 but never shown

Table 2: Would be interesting for the readers whether also PEMT1 and SREBF1 loci are affected? If PEMT1 is affected this should be discussed in light of HDL concentrations.

Table 2: Patients 13/17, deletion Mb?

Table 3: column format

Table 2: Patients 31-35; is it known whether mutations induce loss or gain of function?

Fig. 1 and 2: Units y-axis, please provide significance in Figures.

Line 241/242: What the authors present is far from a ‘complete metabolic profile in patients with SMS’ (therefore also the title should be changed accordingly).

Many of the findings presented her were published more than 20 years ago. Therefore, the authors should clearly communicate the added value in comparison to the study by Smith and colleagues (Ref. 10 of the present manuscript; PMID: 12180145 ).

Author Response

The authors present some laboratory parameters of a relatively small Smith-Magenis syndrome (SMS) cohort (35 patients). SMS is caused by 17p11.2 deletions, encompassing multiple genes including retinoic acid-induced 1 gene (RAI1), or by mutations in the RAI1 gene itself.

The author’s key conclusion is that patients with 17p11.2 deletions (n=30) are prone to develop dyslipidemia while patients with RAI1 mutations (n=5) tend to develop signs of hyperinsulinemia.

The following points (in order of appearence) need attention: Results: lines 148-166 use of integers vs. decimal numbers?

Thanks for the possibility of clarifying that numbers indicated in the text are decimal.

ALT activity mentioned in line 149 but never shown.

ALT is a laboratory datum we had for all SMS patients: unfortunately, it was not analyzed by the statistic assessment. Therefore, following your suggestion, we have omitted the reference to ALT in the text. Thank you for the correction.

Table 2: Would be interesting for the readers whether also PEMT1 and SREBF1 loci are affected? If PEMT1 is affected this should be discussed in light of HDL concentrations.

Thanks for this comment. Indeed, it would have been interesting to discuss PEMT1 and SREBF1 involvement considering HDL concentrations, but - given the heterogeneity of our cohort of SMS patients - this was not considered appropriate at this stage.

Table 2: Patients 13/17, deletion Mb

Thanks, we have added these missing data: for Patient 13: 3.8 Mb and for Patient 17: 9 Mb.

Table 3: column format

The table has been changed, sorry for the mistake.

Table 2: Patients 31-35; is it known whether mutations induce loss or gain of function?

Thank you for the opportunity of pointing out this issue. From the medical literature we have learnt that loss-of-function mechanism is considered responsible for the SMS phenotype. Therefore, given the effect on the transcript, the pathogenic RAI1 variants identified in patients 31-35 are considered to cause RAI1 haploinsufficiency.

Fig. 1 and 2: Units y-axis, please provide significance in Figures.

Figures have been changed: we hope they actually appear clearer to the reader.

Line 241/242: What the authors present is far from a ‘complete metabolic profile in patients with SMS’ (therefore also the title should be changed accordingly).

Thank you, we have omitted the adjective ‘complete’ and corrected the sentence in the discussion, writing that to the best of our knowledge our study is one of the first in the medical literature analysing a wider metabolic profile in patients with SMS.

Many of the findings presented here were published more than 20 years ago. Therefore, the authors should clearly communicate the added value in comparison to the study by Smith and colleagues (Ref. 10 of the present manuscript; PMID: 12180145).

The paper by Smith et al. “Hypercholesterolemia in children with Smith-Magenis syndrome/del (17) (p11.2p11.2)” dealt with the fasting lipid profiles of 49 children between the ages of 0.6 years and 17.6 years with a cytogenetically confirmed diagnosis of SMS. Differently from that, our paper discusses more metabolic variables: of course the 2002 Smith’s paper was a breakthrough in the understanding of this rare and complex syndrome.

Reviewer 2 Report

Cipolla et al. described the metabolic phenotype of a group of 35 patients with Smith-Magenis syndrome, which is a rare genetic disorder associated with heterozygous deletion at chromosome band 17p11.2 region or heterozygous pathogenic variants in the Retinoic Acid-Induced 1 (RAI1) gene. The study provides potentially relevant information about obesity, dyslipidemia, and glucose intolerance/diabetes. However, the lack of control group does not allow the authors to infer if the metabolic features observed in SMS are practically the same as in the general population. This reviewer encourages authors to include a control group. Otherwise, it is not possible to explain the magnitude of the conclusions.

In Line 75, to declare the approval number of the protocol issued by the ethics committee

Table 2 is too long, and it is unclear what to try to show. Then, it should be deleted. Moreover, table 3 shows almost the same information summarized.

Correlations and differences between low and high BMI on metabolic parameters should be corrected by age.

Figure 2 should show the number of cases in each group.

The literature review should be first and show all the collected evidence. Also, the term used and boolean logic should be given.

There are some mispelling and writing issues.

Author Response

Cipolla et al. described the metabolic phenotype of a group of 35 patients with Smith-Magenis syndrome, which is a rare genetic disorder associated with heterozygous deletion at chromosome band 17p11.2 region or heterozygous pathogenic variants in the Retinoic Acid-Induced 1 (RAI1) gene. The study provides potentially relevant information about obesity, dyslipidemia, and glucose intolerance/diabetes. However, the lack of control group does not allow the authors to infer if the metabolic features observed in SMS are practically the same as in the general population. This reviewer encourages authors to include a control group. Otherwise, it is not possible to explain the magnitude of the conclusions.

The lack of a control group population is a limitation of this study, as stated in the final discussion comments. Although the data collected in the SMS population were compared with normative data by age, we thank the reviewer's observation, leading us to modify the conclusions.

In Line 75, to declare the approval number of the protocol issued by the ethics committee.

The approval code from our Ethical Committee is 2105 and the approval date is 5 February 2019 (these data have been added in the text). The Local Ethical Committee approved our study as a part of a larger protocol based on the evaluation of children with disability and nutritional aspects in patients with rare diseases.

Table 2 is too long, and it is unclear what to try to show. Then, it should be deleted.

That table is a record of genetic and metabolic data: we believe that it might be important for readers who want to to check all the information offered in the text.

Moreover, table 3 shows almost the same information summarized.

This table has been redefined with the intent of showing the SMS population divided according to the genetic results, CDC classes (obese/overweight/normal weight) and metabolic data.

Correlations and differences between low and high BMI on metabolic parameters should be corrected by age.

In our study (using correlation analysis and Mann-Whitney U test) we considered BMI-for-age percentiles (“BMI percentiles” in the text) in order to take into account patients’ age. For the use of the Mann-Whitney U test we divided our population into two groups according to lower or higher BMI percentiles (cut-off: 85th percentile). In the text it has been reported that no statistically significant correlation between BMI and total cholesterol, LDL, HDL and triglycerides emerged in our cohort of SMS patients.

Figure 2 should show the number of cases in each group.

Thank you: the figure has now been changed.

The literature review should be first and show all the collected evidence. Also, the term used and boolean logic should be given.

We have defined the modalities of literature review before the discussion (there are 2 supplementary tables clarifying how the review was conducted). Thank you again.

Reviewer 3 Report

Thank you very much for the opportunity to review the article „Metabolic profile of patients with Smith-Magenis syndrome: an observational study and literature review”. This is a very interesting topic. Smith-Magenis syndrome is a genetic disease often accompanied by hypercholesterolemia and obesity. Aim of our study was to describe and characterize the metabolic phenotype of SMS patients, evaluating correlations between body mass index and serum cholesterol, glycated haemoglobin (HbA1c), and basal level of insulin in a cohort of Italian patients with SMS. Understanding the metabolic disorders are crucial to control the disease.

I have a few questions.

The introduction might be improved in details concerning SMS and its relations to metabolic disorders.

In materials and methods section should be clearly stated did the participants take any medications? Were they on special diet?

The results are clearly presented.

In discussion I miss some information about other studies concerning metabolic disorders in different genetic diseases.

Thanks.

Author Response

Thank you very much for the opportunity to review the article "Metabolic profile of patients with Smith-Magenis syndrome: an observational study and literature review”. This is a very interesting topic. Smith-Magenis syndrome is a genetic disease often accompanied by hypercholesterolemia and obesity. Understanding the metabolic disorders are crucial to control the disease. I have a few questions. The introduction might be improved in details concerning SMS and its relations to metabolic disorders.

The relationship of SMS with metabolism is still an open question. In the discussion there is a reference to the paper by Turco et al, evaluating loss of RAI1 and lipid gene expression in a small subset of patients and controls. It is written that “Turco et al. generated and characterized primary cells derived from 4 SMS patients (2 with SMS deletion and 2 carrying RAI1 sequence variants) and 4 control subjects to investigate the molecular processes underlying SMS. By combining transcriptomic and lipidomic analyses, they found that SMS patients had an altered expression of lipid and lysosomal genes, a deregulation of lipid metabolism and a blocked autophagic flux”.

In materials and methods section should be clearly stated did the participants take any medications? Were they on special diet?

We have added this important issue in the proper section, thank you. None of our patients were on a specific diet

The results are clearly presented. In discussion I miss some information about other studies concerning metabolic disorders in different genetic diseases.

We have added some references dealing with other genetic conditions in which lipid metabolism might result impaired.

Round 2

Reviewer 1 Report

All points are adequately addressed.

Author Response

Thank you for appreciating the changes and amendments made

Reviewer 2 Report

The authors have addressed the reviewer's comments.

The title of 3.3 should be “Comparisons of metabolic parameters between patients with low and high BMI percentiles.”

It is not clear whether the review of medical literature is an individual section or part of the discussion. If the editor agrees, the authors may entitle the section as “Discussion and review of medical literature”. Firstly, summarize the evidence of the literature and then add the new findings of the present study.

Author Response

The authors have addressed the reviewer's comments. The title of 3.3 should be “Comparisons of metabolic parameters between patients with low and high BMI percentiles.”

Thank you for the opportunity of clarifying that the section was dedicated to the comparison of metabolic parameters with BMI, but not only to BMI. The comparison was also made with patients’ age and gender. We felt appropriate to modify the 3.3. section writing a more generic title”Application of the Mann Whitney U-test”. Hope you will understand this choice.

It is not clear whether the review of medical literature is an individual section or part of the discussion. If the editor agrees, the authors may entitle the section as “Discussion and review of medical literature”. Firstly, summarize the evidence of the literature and then add the new findings of the present study.

Thank you: in the specific section, named “3.5. Review of medical literature”, we explained the modalities of our literature search. The whole discussion is a mixture of considerations in which the relationship between Smith-Magenis syndrome and metabolic dysregulation has been explored. From the paper by Lacaria et al. to that by Shimano et al. many findings have been sequentially compared to our results.
